# ATEX-Certified, FPGA-Based Three-Channel Quantum Cascade Laser Sensor for Sulfur Species Detection in Petrochemical Process Streams

**DOI:** 10.3390/s25030635

**Published:** 2025-01-22

**Authors:** Harald Moser, Johannes Paul Waclawek, Walter Pölz, Bernhard Lendl

**Affiliations:** 1Institute of Chemical Technologies and Analytics, TU Wien, Getreidemarkt 9/164, 1060 Vienna, Austria; johannes.waclawek@tuwien.ac.at (J.P.W.); bernhard.lendl@tuwien.ac.at (B.L.); 2OMV AG, Mannswörther Straße 28, 2320 Schwechat, Austria; walter.poelz@omv.com

**Keywords:** infrared laser spectroscopy, quantum cascade lasers, laser sensor, hydrogen sulfide, carbonyl sulfide, methyl mercaptan, methane, HDS, FCC, process monitoring

## Abstract

In this work, a highly sensitive, selective, and industrially compatible gas sensor prototype is presented. The sensor utilizes three distributed-feedback quantum cascade lasers (DFB-QCLs), employing wavelength modulation spectroscopy (WMS) for the detection of hydrogen sulfide (H_2_S), methane (CH_4_), methyl mercaptan (CH_3_SH), and carbonyl sulfide (COS) in the spectral regions of 8.0 µm, 7.5 µm, and 4.9 µm, respectively. In addition, field-programmable gate array (FPGA) hardware is used for real-time signal generation, laser driving, signal processing, and handling industrial communication protocols. To comply with on-site safety standards, the QCL sensor prototype is housed in an industrial-grade enclosure and equipped with the necessary safety features to ensure certified operation under ATEX/IECEx regulations for hazardous and explosive environments. The system integrates an automated gas sampling and conditioning module, alongside a purge and pressurization system, with intrinsic safety electronic components, thereby enabling reliable explosion prevention and malfunction protection. Detection limits of approximately 0.3 ppmv for H_2_S, 60 ppbv for CH_3_SH, and 5 ppbv for COS are demonstrated. Noise-equivalent absorption sensitivity (NEAS) levels for H_2_S, CH_3_SH, and COS were determined to be 5.93 × 10^−9^, 4.65 × 10^−9^, and 5.24 × 10^−10^ cm^−1^ Hz^−1/2^. The suitability of the sensor prototype for simultaneous sulfur species monitoring is demonstrated in process streams of a hydrodesulphurization (HDS) and fluid catalytic cracking (FCC) unit at the project’s industrial partner, OMV AG.

## 1. Introduction

The precise and selective detection of gaseous sulfur compounds, particularly hydrogen sulfide (H_2_S), carbonyl sulfide (COS), and methyl mercaptan (CH3SH), is of paramount importance in a variety of petrochemical processes, including production monitoring, quality control, and personal safety applications. Given the prevalent presence of sulfur species in these processes and their adverse effects on both process efficiency and product integrity, it is critical to maintain tight control over their concentrations in the sub-ppmv range for H_2_S and CH_3_SH. Furthermore, the stringent detection sensitivity of approximately 50 ppbv for COS quantification adds another layer of complexity.

Despite the availability of various online monitoring methods for gaseous sulfur species, their reliable, quantitative, and selective detection continues to pose significant challenges in the field of chemical sensors [1,2,3,4]. Different analytical approaches applicable to the quantification of gaseous sulfur species with the industrial and biotechnological focus on H_2_S encompass spectroscopic methods, followed by fluorescence-based assays, and electrochemical, colorimetric sensor, and (gas) chromatographic methods [5,6,7].

Hydrodesulfurization (HDS) and fluid catalytic cracking (FCC) are critical processes in contemporary petroleum refining. These technologies have gained heightened attention due to increasingly rigorous environmental regulations aimed at reducing the sulfur content in transportation fuels, including gasoline, diesel, and jet fuel, as well as in olefinic gases and other key petrochemical products. Firstly, sulfur-containing impurities in hydrocarbon fuels act as potent catalyst poisons, inhibiting the utilization of untreated crude feedstocks in subsequent chemical processes. Secondly, the combustion of these impurities leads to the release of sulfur oxides (SO_X_), which have significant environmental consequences including their role as precursors to acid rain formation [8].

Since its initial operational demonstration in 1994 [9], the quantum cascade laser (QCL) has evolved into a highly effective and reliable spectroscopic light source, operating in the mid-infrared (MIR) and terahertz regions. This advancement has enabled the sensitive detection of molecular species based on their fundamental vibrational bands, thereby establishing laser-based absorption spectroscopy as a powerful tool for industrial gas sensing [10,11,12,13].

Efforts to maximize gain bandwidth have highlighted the importance of broadly tunable and single-mode emission QCLs for applications in chemical, physical, biological, and atmospheric sensing. These lasers enable the simultaneous measurement of multiple molecular absorption features within the MIR spectral range, enhancing their utility in precise and comprehensive spectroscopic analyses [11,14,15,16].

For a clear identification of the target molecules, laser specific spectral windows with strong absorption features need to be chosen, whereas the interference of absorption lines from matrix molecules is minimized. Eventually, the choice of the spectral window is also dependent on the availability of an adequate radiation source. The spectroscopic detection of multiple species and transitions requires devices capable of providing a high spectral resolution and a spectral coverage wide enough to fully encompass absorption features. Mode-hop-free external cavity QCLs (MHF EC-QCLs) with tuning ranges in the order of ~100 cm^−1^ are commercially available and their usage has been reported [17], but the long scan times and limited modulation performance in terms of modulation depths could pose a problem for operational scenarios where fast concentration feedback is needed. Under these circumstances, the preferred sensor specific transition to multi-species capabilities usually involves multiplexing individual laser sources together with well-matched analyte and spectral transition pairings within the limited bandwidth. In this regard, multi-species sensors with multiplexed near-infrared (NIR) laser sources [6] and NIR-MIR hybrid approaches [7] are worthwhile to point out.

Distributed-feedback (DFB)-type QCLs satisfy the stringent requirements for single-mode emission and superior wavelength stability, making them ideal for industrial trace gas sensing applications. Commercially available DFB-QCLs are typically configured as edge-emitting ridge lasers and are housed in standardized, thermo-electrically (TE) stabilized high heat load (HHL) semiconductor packages. These lasers are tunable through modulation of the injection current and/or adjustments to the temperature of the gain medium. However, the tuning range is typically limited to only a few wavenumbers, restricting the ability to target more than one or two analytes spectroscopically with a given DFB-QCL.

Various approaches to QCL-based quantitative multi-species gas-phase spectroscopy have been explored, with the technical details recently reviewed [16]. These methods include cavity-enhanced absorption spectroscopy [18], quartz-enhanced photoacoustic spectroscopy [19], and open-path setups [20], all of which have been demonstrated to be successful in industrial and environmental monitoring applications.

The “gold standard” for QCL-based trace gas measurements in the MIR spectral range involves absorbance measurements using multipass reflection cells [21]. While alternative cell designs have been tested with success [22], the Herriott cell remains the standard configuration due to its ability to provide a long effective optical interaction pathlength, often extending to several tens of meters [23].

When combined with phase-sensitive detection techniques, such as wavelength modulation spectroscopy (WMS), the influence of 1/*f* electronic noise can be significantly reduced, enabling high detection sensitivities [24,25,26]. WMS involves the application of kHz- to MHz-frequency sinusoidal modulation to the laser radiation [8]. Following an interaction with the target gas in the gas cell, the modulated signal is processed using a lock-in amplifier (LIA) for demodulation. The modulation process effectively shifts the spectroscopic information to higher frequencies, thereby mitigating the impact of 1/*f* noise, which is most pronounced at lower frequencies [21]. The acquired data can be analyzed by demodulating the signal at the harmonics of the modulation frequency. In particular, demodulation at the second-harmonic frequency (2*f*) efficiently suppresses baseline offsets and slopes. However, the extracted WMS signals are susceptible to fluctuations in laser power. To address this issue, a widely adopted approach involves normalizing the 2*f*-signal by the first-harmonic (1*f*) component, a technique coined as the WMS-2*f*/1*f* method [8].

Compared to traditional tunable diode laser absorption spectroscopy (TDLAS), the extraction and fitting of spectral information in QCL-based WMS are more complex due to the pronounced non-linear modulation characteristics [9]. WMS with accurate spectral fitting routines and strategies for rapid inversion of gas concentration requires meticulous control of the laser operational parameters to account for the effects of modulation and to ensure the acquisition of reliable absorption spectra. From this perspective, multiple linear regression algorithms with the use of known reference spectra, derived from calibrated gas cylinders, are employed [10,11]. WMS detection with direct absorption spectroscopy correction based dual-spectroscopy technique is reported [12], and a self-calibrated 2f/1f wavelength modulation spectroscopy approach are highlighted.

Recent advances in digital signal measurement and conditioning technologies are making cutting-edge signal processing methods both possible and more accessible to laser-based spectroscopy applications. In particular, field-programmable gate arrays (FPGAs) offer users a programmable chip capable of real-time data manipulation. FPGAs contain large resources of programmable logic blocks and block random access memory (BRAM) which, when appropriately “synthesized”, are configured to perform complex mathematical manipulations at clock speeds of 40 MHz up to 1 GHz.

In this context, a selective and sensitive sulfur species sensor able to discriminate and quantify H_2_S, CH_3_SH, and COS among hydrogen and methane (CH_4_) for the HDS process, and hydrocarbon matrices (propene, C_3_H_6_) for the FCC process, respectively, with a fast response time is of utmost process analytical value in order to maintain the HDS or FCC units under optimum operational parameters [27,28].

## 2. Materials and Methods

### 2.1. Design Consideration of the Sensor and Optical Principles

The presented sensor architecture is the current stage of optimization efforts of previously reported works of the authors: In particular, compared to the first sensor generation [29], the number of laser sources and thus accessible target analytes was increased in the second design iteration [30]. Moreover, a permanent reference channel was installed, and the sensor footprint was further reduced. The current sensor generation is displaying a totally revised and novel redesign of the laser source combination module, thus greatly enhancing the modularity of the optical train. Furthermore, the sensor is now equipped with field-programmable gate array (FPGA) hardware, used for real-time signal generation, laser driving, signal processing, and handling industrial communication protocols.

The prototype is based on scanned wavelength modulation spectroscopy (WMS), employing three time-division multiplexed laser sources. The scanned-wavelength WMS strategy involves adding a fast (kHz range) sinusoidal modulation (at frequency *f*) to a slow (Hz range), arbitrary periodic ramp function (e.g., sawtooth, sine, triangle, etc.) signal to the laser injection current. In WMS, the absorption information is thus shifted to higher frequencies with the benefit of sufficient isolation from 1/*f* laser noise and low-frequency fluctuations, thus ultimately resulting in an increased signal-to-noise ratio (SNR). In this fashion, the laser is tuned over an absorption line and attenuated by the presence of the analyte. After passing the gas measurement cell, the light impinges upon a photodetector, and the resulting voltage signal is processed via synchronous demodulation (lock-in process) in order to extract the signal components at the fundamental modulation frequency (*f*) and its harmonics (1*f*, 2*f*, 3*f*, etc.). To achieve sufficient separation of the harmonics for more effective filtering, the modulation frequency is usually more than two orders of magnitudes greater than the scanning rate. The detection bandwidth (response time) will be limited by the slower scanning frequency, similarly to a scanned direct absorption scheme.

### 2.2. Laser Characterization and Selection of Spectral Windows

Due to the spectroscopic properties of the analytes of interest, their absorption lines are clearly separated in the range between 4 and 8 µm. The presence of strongly absorbing matrix molecules in the process gas, such as CH_4_, C_3_H_6_, and other hydrocarbons, presupposes a precise wavelength selection of the laser sources.

Laser-specific spectral windows are carefully chosen in order to ensure clear identification of the target molecules by targeting strong absorption features, while minimizing the interference from absorption lines of matrix molecules.

The QCL sources used in this sulfur species sensor are commercially available continuous wave (cw) DFB-QCLs and mounted in a high heat load (HHL) package. The laser parameters are stated in Table 1. Each device was characterized with a high-resolution FTIR spectrometer (Vertex 80v, Bruker, Berlin, Germany), whereas spectra were recorded at different gain element temperatures and laser currents.

The individual absorption spectra of the target analytes within the coarse tuning ranges of the employed QCLs are shown in Figure 1. In addition, absorption spectra of the four most important hydrocarbons, which are the major matrix constituent encountered in petrochemical process streams, as outlined in the section above, are plotted as well.

For the selection of the most suitable range for the detailed and optimum spectral detection of each target analyte, reference spectra at a reduced pressure of 100 mbar were calculated [31] based on the HITRAN [32] database. In addition, spectral simulations of available reference gas cell fillings (i.e., CH_4_ for the H_2_S and CH_3_SH channels and CO for the COS channel) for implemented wavelength tracking are supplemented. The emitted wavenumber and attributed power of each QCL channel, depending on the operation parameters, are plotted in Figure 2a–c, and spectra of the target analytes reference cell analytes, along with the designated operational area, are shown. As indicated by the shaded area of each laser channel, the designated operational spectral window can be reached with moderate TEC current settings (i.e., operational points are within 20–30 °C). In addition, the high laser optical output power can still be leveraged for the usage in the beam combination module.

## 3. Instrumentation

### 3.1. Optical Layout

The sensor architecture uses a viable alternative to the popular beam splitter train or dichroic ladder configuration, as encountered in many spectroscopic applications in order to combine multiple individual laser emitters. The alternative beam combination method utilizes a geometric arrangement of individual narrowband spectral filter (NBSF) elements on the perimeter of a regular (i.e., equilateral and equiangular) pentagon. The applicability of this beam combiner within a similar optical layout together with a detailed evaluation of beam combination efficiencies and beam quality parameters (M^2^) as well as a comparison with a previous setup designed with conventional beamsplitter arrangements was recently reported [33].

The individual filters (H_2_S:FB8000-500, CH_3_SH: FB7500–500, COS: FB4750-500, Thorlabs, Dortmund, Germany) transmit the specified wavelength band (T > 0.8), while effectively reflecting (R >> 0.95) out-of-band emission and blocking unwanted radiation. In this fashion, all three individual emitters are co-aligned with two gold mirrors (PF-10-M03, Thorlabs, Newton, NJ, USA) in a Z-fold configuration. The resulting collimated and multiplexed laser beam is split into a reference and signal path (CaF_2_ window, 95:5 splitting ratio, Thorlabs) before being spatially filtered, mode matched, and coupled into an astigmatic 76 m Herriott multipass cell (AMAC76-LW, Aerodyne Inc., Billerica, MA, USA). The exiting beam is subsequently focused onto an optically immersed, thermoelectrically cooled mercury–cadmium–telluride (MCT) detector (PVI-4TE-12, Vigo Systems, Ożarów Mazowiecki, Poland). The reference path is equipped with a second MCT detector (PCI-4TE-9, Vigo Systems) as well as with methane (CH_4_), hydrogen sulfide (H_2_S), and carbon monoxide (CO) reference gas cells (Wavelength References, Corvallis, OR, USA) for accurate wavelength calibration and laser drift compensation. The detector signals are digitized, demodulated, and further processed using a self-developed multi-channel filtering and lock-in amplifier chain implemented in FPGA hardware (NI 7856R, National Instruments, Austin, TX, USA). The optical layout of the triple-QCL sensor is outlined in Figure 3.

### 3.2. Signal Generation and Data Acquisition

A field-programmable gate array (FPGA)-based, multifunction reconfigurable input/output device based on a Kintex-7 160T (NI 7856R, National Instruments) was used for onboard processing and direct control over the driving and acquisition signals for complete flexibility of system timing and synchronization. The control signal for the individual lasers consists of a modified low-frequency sawtooth ramp with a superimposed high-frequency modulation waveform. The rather slow sawtooth function is responsible for tuning the emitted wavelength over the absorption lines of interest, whereas the high-frequency waveform is responsible for the superimposed intensity and wavelength modulation.

All implemented FPGA cores run at a 80 MHz base clock including the numeric oscillator, direct digital synthesis (DDS), and digital input and output lines. However, the sampling frequency of the analog-to-digital converters (ADCs) and digital-to-analog converters (DACs) is clocked at 1 MHz.

The internal flow diagram of the digital and analog pathways is schematically represented for one channel in Figure 4. The direct digital synthesis (DDS) core generates a digital representation of the low frequency ramp waveform and the high-frequency modulation waveform (A). The reference ramp and modulation signals are generated by employing numeric oscillators with an internal 32-bit phase accumulator. After adding stages (B), the resulting digital waveforms are sent to the designated output channels of the DACs (C).

The detector signals are digitized (C) and de-glitched with an 11-tap median filter (E). The demodulation can be chosen to follow the traditional dual phase lock-in pathway with mixing of the input signal with a reference sine and cosine (G), followed by low-pass filtering (LPF) of the resulting in-phase and out-of-phase signal components (I) and generating the proper information (J).

In addition, a novel approach of a high-bandwidth synchronous demodulation technique which utilizes phase cancelation to substantially improve the performance of the digital lock-in amplification process [34] is implemented. Here, after median filtering (E), a phase shift of 90° is introduced to the input signal by employing a self-designed FIR 83-tap Hilbert transform filter (F) and subjecting it to the mixing procedure. The resulting four-fold signals from the demodulation stage (G) are mathematically combined (H) in order to nearly suppress the unwanted frequency component of twice the modulation frequency (2ω_0_) via phase cancelation, thus considerably reducing the amount of needed filter coefficients of the LPF stages (I). The LPF filter stages are realized as a decimating cascaded integrator comb (CIC) section [35], followed by a linear-phase, tapped-delay, multi-stage, moving average (MA) filter, in order to compensate for the non-flat, sin(x)/x-like magnitude response of the passband of the preceding CIC stages.

The resulting spectra of the chosen harmonics are transferred to a measurement computer for further post-processing and storage.

### 3.3. Laser Driving and Peripheral Components

As each QCL requires its own high-precision current driver (QCL Series OEM, Wavelength Electronics, Bozeman, MT, USA) for matched power delivery, the control signal is specific and different in terms of threshold, low and high level. Each laser is protected with a dedicated fuse and an ESD absorber (Lasorb LA44-2000, Pangolin Systems, Sanford, FL, USA). Time-division multiplexing is performed with different modulation frequencies (typically 5–15 kHz) in order to differentiate the currently active laser in the FPGA hardware and PC software (LabVIEW v2020, National Instruments, Austin, TX, USA).

The sensor requires the precise and accurate temperature control of the laser gain chips, realized by a dedicated TEC controller (TEC 1091, Meerstetter, Rubigen, Switzerland) for each HHL package. An additional high-load TEC controller (TEC 1089, Meerstetter, Rubigen, Switzerland) is installed to stabilize the temperature of the aluminum heatsink plate where all three HHL-DFB-QCLs are mounted.

The pressure in the multipass gas cell is maintained via a four-stage diaphragm vacuum pump (MV 10 NT Vario, Vacuubrand, Wertheim, Germany) and regulated with a pressure controller (GSP-C5SA, Vögtlin, Muttenz, Switzerland) and typically set to a pressure range of 30–100 mbar and flow range of 0.1–2 L/min, depending on the application and matrix composition.

### 3.4. ATEX Sensor Architecture and Process Implementation

In order to meet on-site safety regulations, the three-channel QCL sensor platform is installed in a purged and pressurized (EXp) industry enclosure (KE 9408.600, Rittal, Wetzlar, Germany) and equipped with the required safety infrastructure (EPV 5500, P&F, Mannheim, Germany) allowing a certified operation under ATEX/IECEx regulations for hazardous and explosive environments (Figure 5). This method of explosion protection is the technique of choice which ensures that the differential pressure of a protective gas (N_2_, instrument air) inside an EXp enclosure is sufficient to prevent the ingress of a flammable gas, vapor, dust, and fiber, and prevent a possible ignition [36].

In detail, the sensor integrates a fully automated gas sampling and conditioning system (flow rate and pressure) with a dual compartment (inner and outer shell) and dual source purge and pressurization system (N_2_ and instrument air), along with built-in safety electronic devices, providing intrinsic explosion prevention and malfunction protection.

Prior to normal operation, the startup procedure for safe and ATEX-compliant operation involves purging and subsequently maintaining a constant pressure level of inert gas (N_2_) in the inner aluminum compartment, housing the measurement cell. Additionally, the EXp instrument enclosure is purged with instrument air. Leakage losses in both the pressurized inner compartment and enclosure are compensated so that the designated pressure levels remain constant. In case of sudden pressure drops in either or both compartments, immediate power shutoff is commenced.

## 4. Results and Discussion

### 4.1. Wavenumber Tracking and Interpolation

The reference path arm (Figure 3), equipped with the matching reference cell for each laser, is used for spectral and power monitoring. The digitized harmonic spectra are constantly evaluated and serve as input for a wavenumber tracking procedure. Wavenumber tracking and subsequent interpolation can be successfully implemented after peak detection and peak identification via the database of the recorded 2*f*-WMS reference cell spectrum are performed. In a first step, the lock-in time position (spectral index) of the detected peaks is extracted and correlated with the matching wavenumber positions of the corresponding database peaks. The resulting spectral index–wavenumber pairs are least squares fitted with appropriate low-order polynomials (*n* < 3). In a last step, the original linear spectral index vector is now exchanged with the non-linear tuning fit wavenumber vector. The procedure is outlined in Figure 6.

In order to compare the effects of the wavenumber tracking with its benefits of absolute spectral positioning and stabilized wavenumber output of the sensor, the position differences in the first and last detected peaks in each scan in terms of the spectral index (ΔPP_SI_, free-running operation) and wavenumber (ΔPP_ν_, tracked and interpolated operation) are recorded for a duration of 4000 s. Allan–Werle deviation plots of the position differences are shown in Figure 7 for the CH_3_SH QCL in free-running (left) and tracked operation (right).

While QCL2 in free-running operation begins to experience drift after ~60 s (as indicated by the minimum in Figure 7, left), the tracked operation drastically improves stability. In comparison, Allan–Werle deviation plots for the H_2_S QCL are presented in free-running (left) and tracked operation (right). While QCL1 in free-running operation begins to experience considerable drift only after ~16 s (as indicated by the minimum in Figure 8, left), the tracked operation again drastically improves stability.

### 4.2. Limit of Detection (LoD)

The characterization, metrological assessment, and validation of the spectrometer at representative conditions in the laboratory are highly demanding, given the wide span of target analyte concentration and vastly varying matrix composition of real process gases.

The first step towards a successful multi-analyte detection was the evaluation of selectivity and sensitivity. Quantitative measurements of H_2_S, CH_3_SH, and COS, were performed using calibration gas mixtures in nitrogen (N_2_), methane (CH_4_), and propene (C_3_H_6_) matrices to investigate the sensitivity and linear response of the 2*f*-WMS-based triple-QCL sensor system. Different target analyte concentration levels were prepared by dilution from these standardized gas bottles with their respective main matrix component using a mass flow and pressure-controlled in-house-developed gas handling system. The sensor was operated with the multipass cell at room temperature (22 ± 2.5 °C), flow rates varying from 0.1 to 1.0 L/min, and a total pressure ranging from 15 mbar to 100 mbar, enabling the spectral resolution of the ro-vibrational bands of both the analyte and matrix molecules.

Each concentration step with equidistant spacing regarding the total concentration span was measured 10 times, and the resulting averages are plotted as a function of concentration (refer to Figure 9). Good linearity between 2*f*-WMS signal amplitudes and target analyte concentrations are observed for all three QCL channels.

The corresponding limits of detection (LOD) were assessed according to DIN 32645 at three times the standard deviation (3σ) of the intercept divided by the slope of the calibration curve and resulted in ~0.3 ppmv for H_2_S, ~60 ppbv for CH_3_SH, and ~5 ppbv for COS, respectively.

### 4.3. Noise-Equivalent Absorption Sensitivity (NEAS)

In addition to the detection limits, a more detailed metric for comparing spectroscopic instruments is the noise-equivalent absorption sensitivity (NEAS), defined as the minimum detectable absorption, scaled by both the pathlength and the noise-equivalent detection bandwidth [16,37]:(1)$$NEAS=\left(\frac{\Delta I}{I}\right)\left(\frac{1}{L\sqrt{BW/N}}\right)$$

Here, the first term, Δ*I*/*I*, is the 1σ value of the limiting noise level in the spectrum, normalized by the total intensity *I*. The second term is composed of *L* as the optical pathlength, *BW* as the detection bandwidth, and *N* as the number of measurement averages.

For the NEAS determination, 2*f*-WMS spectra of 25 ppmv H_2_S, 50 ppmv CH_3_SH, and 1.0 ppmv COS were recorded with a lock-in time constant of 2 ms, a filter slope (roll-off) of 24 dB/octave, and within a single shot measurement (no averages), which resulted in a noise-equivalent bandwidth of the low-pass filter of 39 Hz. The results are plotted in Figure 10 and the calculated values of the NEAS for each analyte are listed in Table 2.

Upon closer inspection, a relatively high fringe background can be observed, especially for the H_2_S and CH_3_SH channels. Unfortunately, the demanded ATEX compliance under an EXp operation requires the use of a compartmented approach; thus, additional optical components (i.e., optical window as sketched in Figure 5) have to be installed. As a consequence, and despite the use of a wedged and broadband-transmitting ZnSe window (WW71050, Thorlabs), the fringe contrast deteriorates the optical signal. Interestingly, the effect seems to be more pronounced at 7.5 µm (CH_3_SH channel) than at 8 µm (H_2_S channel), and appears to be absent at 4.7 µm (COS channel).

### 4.4. Long-Term Stability—Allan–Werle Variance Analysis

The temporal stability of the sensor was evaluated by measuring a constant concentration of the designated target analytes on the according QCL channels for roughly 7 h (QCL1: 40 ppmv H_2_S; QCL2: 25 ppmv CH_3_SH; QCL3: 200 ppbv COS). The Allan–Werle variance analysis (Figure 11) retrieved an optimum integration time between 80 and 100 s. These results are well aligned with reported findings [4,38], and it is assumed that thermal drifts, influenced by the ATEX compartment and purge air parameters (flow rate and temperature variations) limit the maximum stable operation. Here, it has to be noted that each laser channel was operated individually for the course of the full recording. After setting the desired concentration for each target analyte, the operational point of each laser was line-locked with the help of the employed reference cells, as marked by the evaluated SNR position in Figure 10.

### 4.5. On-Site Process Spectra

The results from the first online measurements under fully engaged ATEX operation from a sampled hydrodesulphurization unit (HDS3) at the project’s partner, OMV AG, are depicted in Figure 12. The exemplary data were recorded during transient and thus unstable unit operational conditions after weeks of shut-down due to maintenance (turn-around). The absolute values of concentration scales and additional plant parameters, namely current temperature and pressure levels, are omitted due to company regulations. However, the general unstable operational conditions of the fundamental process are interesting for sensor evaluation as dynamic concentration fluctuations are expected and welcomed. The long dead time until the process gas finally arrives at the sensor inlet is due to the mandatory N_2_ purge of the sampling pipes before the sensor power-on. A representative snap-shot at t = 2500 s (marked by arrows) of the spectral composition of each QCL channel with the according accessible analytes is visualized in Panel B. In general, the concentration time series are evaluated at the same spectral positions as used for the SNR positions of Figure 10.

## 5. Summary and Conclusions

A prototype MIR optical gas sensor using wavelength modulation spectroscopy with second-harmonic detection (2*f*-WMS) and three continuous-wave distributed-feedback quantum cascade lasers (CW-DFB-QCLs) was developed for the fast, sensitive, selective, and simultaneous detection of H_2_S, CH_3_SH, and COS for sulfur species quantification in petrochemical applications. Detection limits of ~0.3 ppmv for H_2_S, ~60 ppbv for CH_3_SH, and ~5 ppbv for COS could be achieved.

The sensor was installed in an industrial rack equipped with ATEX safety infrastructure to meet petrochemical safety regulations, and it was tested on-site at OMV AG.

The challenge for industrial QCL implementation lies in balancing tailored applications with costs and benefits. While the MIR spectral region is preferred for its stronger chemical cross-sections, it requires expensive optical components. However, QCL technology is expected to penetrate industrial markets where no other solution exists for process gas detection. Key process analytical advantages of the developed sensor prototype include its high selectivity, sub-ppmv sensitivity, and rapid response time, enabling the continuous and direct measurement of process gas streams without the need for sample pre-treatment.

Ongoing research regarding versatile spectroscopic sources is expected to expand QCL sensor capabilities, using designs like multiple DFB chips, RCSE arrays, and Vernier-effect QCLs for broader spectral coverage and the subsequent detection of more analytes.

Integrating laser sources and detectors into a single device could enable the development of miniaturized, multi-species gas sensors. In addition, focusing not only on direct absorption techniques but also further extending to photothermal- [39] and dispersion-based measurement capabilities [40] will help close the gap on established process GC analyzers.

The further development of advanced safety features to achieve ATEX certification with minimal impact on optical performance is necessary to fully take advantage of the MIR spectroscopic benefits while ensuring compliance with petrochemical regulations.

## Figures and Tables

**Figure 1 sensors-25-00635-f001:**
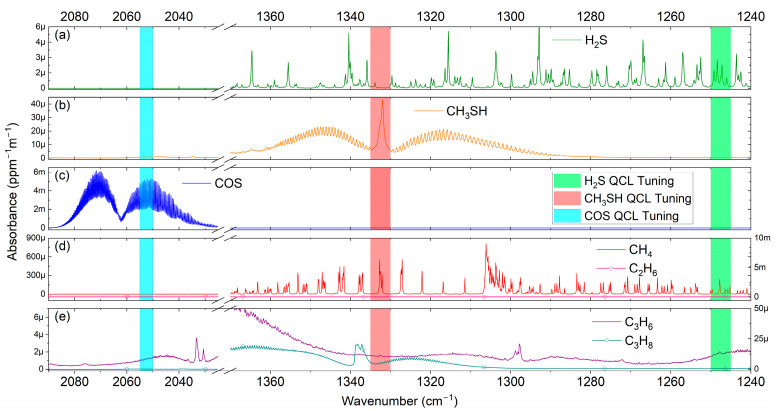
Absorption spectra of the target analytes (1 ppm-m, 25 °C, 1 bar) along with the tuning range of the employed QCLs. (**a**) H_2_S, (**b**) CH_3_SH, and (**c**) COS. Absorption spectra of the most important hydrocarbon matrices are plotted for CH_4_ and C_2_H_6_ in (**d**) and for C_3_H_6_ and C_3_H_8_ in (**e**). All spectra are taken from the PNNL database.

**Figure 2 sensors-25-00635-f002:**
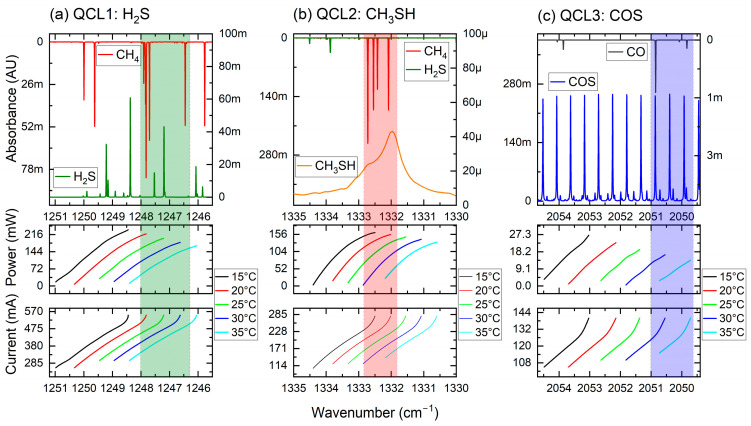
Characterization of the cw-QCLs. Operational current, emitted wavenumber, and attributed optical power are plotted in the lower parts of the figure. In addition, absorption spectra of the target and reference cell analytes, along with the designated operational area (shaded), are plotted in the top parts.

**Figure 3 sensors-25-00635-f003:**
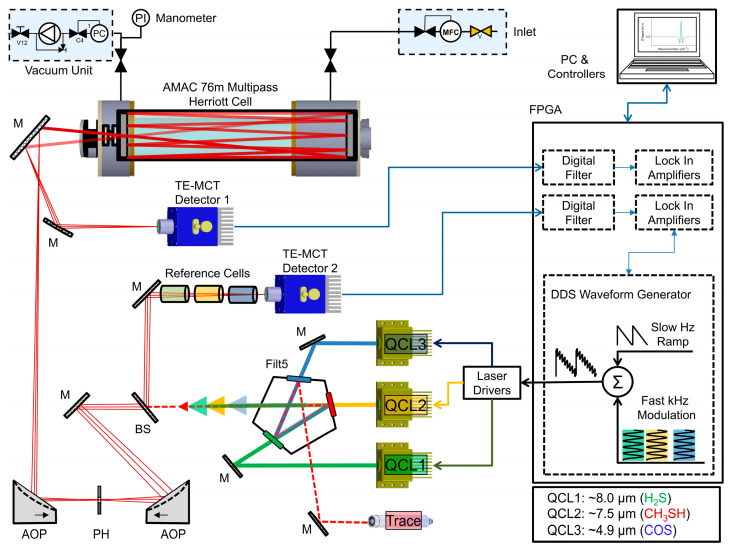
Optical and peripheral layout of the triple-QCL sensor. BS: beamsplitter; M: mirror; AOP: off-axis parabolic mirror; PH: pinhole.

**Figure 4 sensors-25-00635-f004:**
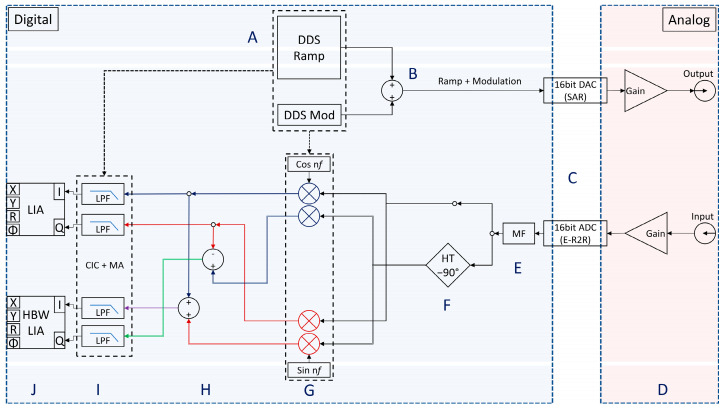
Schematic of the FPGA implementation with different functional submodules: Direct digital waveform generation (**A**,**B**), DAC output and ADC input (**C**,**D**), digital filtering (**E**,**F**), digital lock-in amplifier stages (**G**,**H**) and subsequent low pass filtering stages (**I**,**J**).

**Figure 5 sensors-25-00635-f005:**
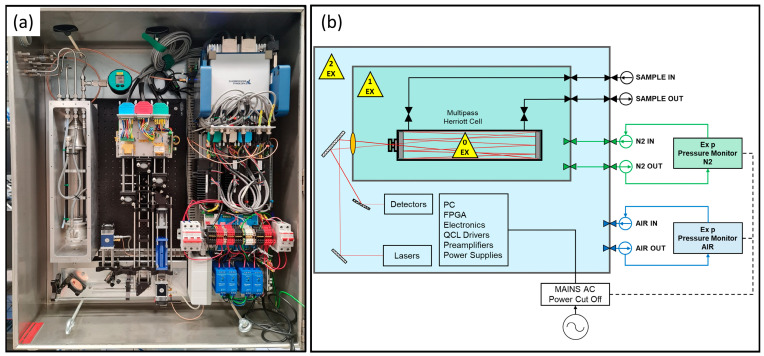
(**a**) Fully assembled sensor and peripheral components. (**b**) Piping, instrumentation, and ATEX safety flow diagram.

**Figure 6 sensors-25-00635-f006:**
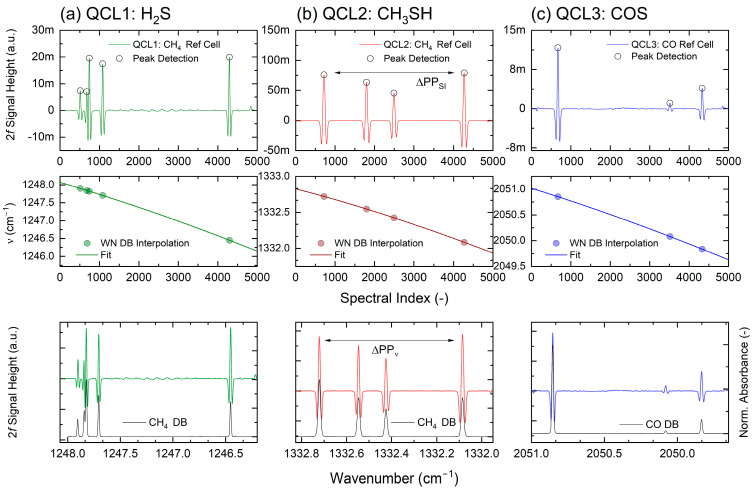
Outline of the wavenumber tracking and interpolation steps of the three QCL channels.

**Figure 7 sensors-25-00635-f007:**
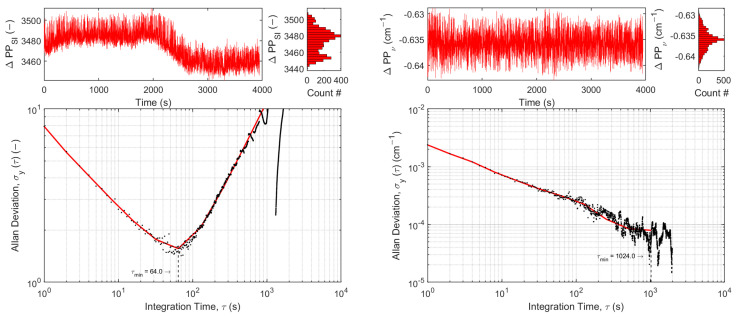
Allan–Werle deviation plots of the position differences in the first and last detected peaks in terms of spectral index (ΔPP_SI_,) for the QCL2: CH_3_SH in free-running operation (**left**) and the position differences in terms of wavenumber (ΔPP_ν_,) for tracked operation (**right**).

**Figure 8 sensors-25-00635-f008:**
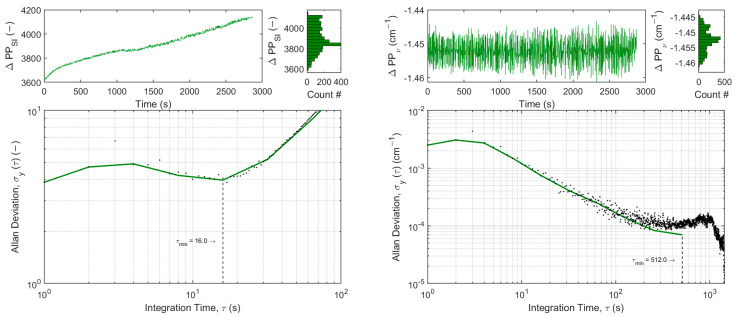
Allan–Werle deviation plots of the position differences in the first and last detected peaks in terms of spectral index (ΔPP_SI_,) for the QCL1: H_2_S in free-running operation (**left**) and the position differences in terms of wavenumber (ΔPP_ν_,) for tracked operation (**right**).

**Figure 9 sensors-25-00635-f009:**
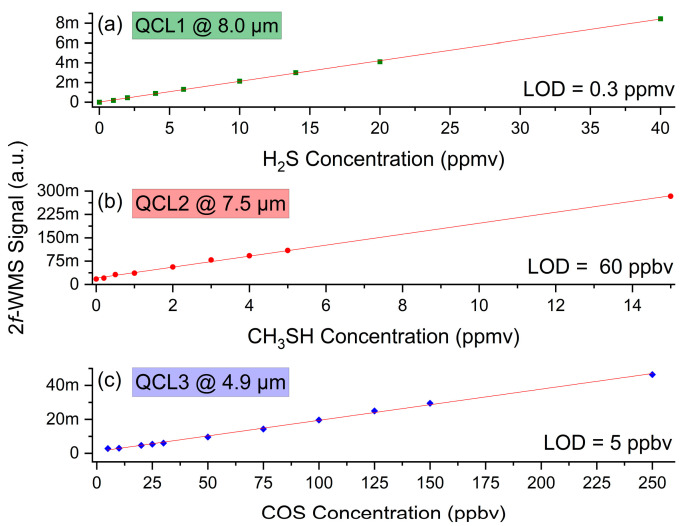
Calibration curves of 0–40 ppmv H_2_S, 0–15 ppmv CH_3_SH, and 0-250 ppbv COS in N_2_. The calculated LOD (3σ) are 0.3 ppmv for H_2_S, 60 ppbv for CH_3_SH, and 5ppbv for COS.

**Figure 10 sensors-25-00635-f010:**
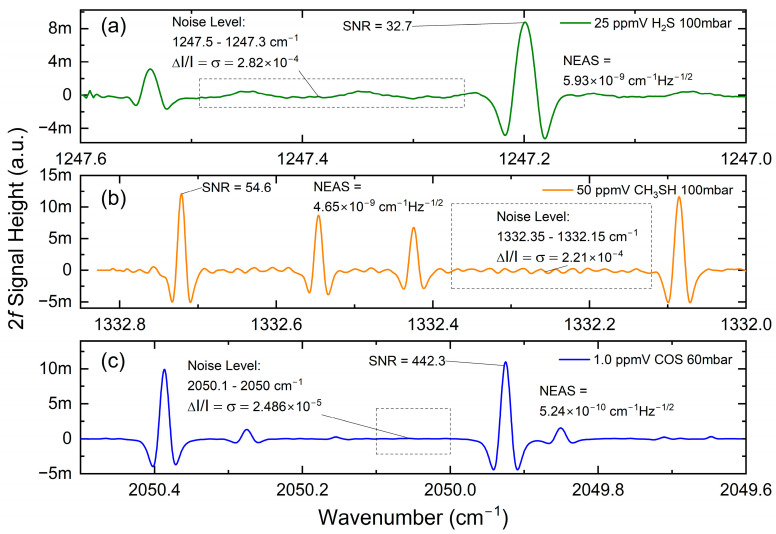
Determination of the noise-equivalent absorption sensitivity (NEAS) of (**a**) H_2_S, (**b**) CH_3_SH, and (**c**) COS.

**Figure 11 sensors-25-00635-f011:**
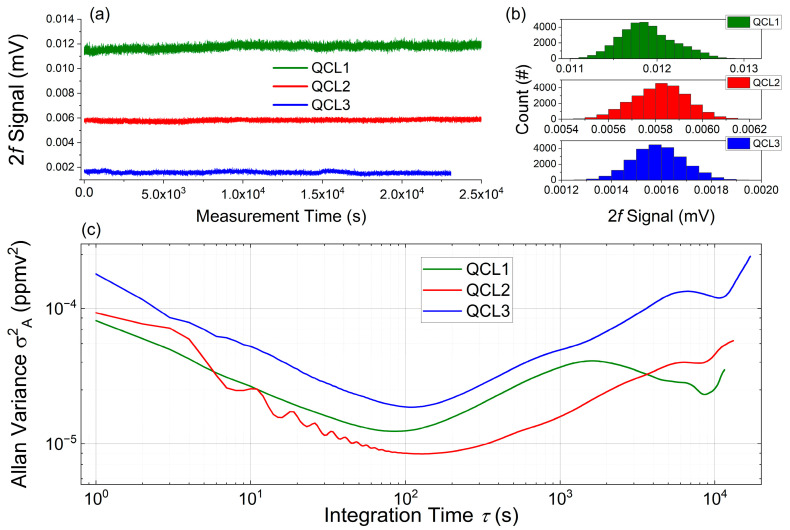
(**a**) Time series measurements of constant analyte concentrations for over 7 h. (**b**) Histograms of the time series measurements. (**c**) Determination of the Allan variance as function of integration time: QCL1: H_2_S (green); QCL2: CH_3_SH (red); QCL3: COS (blue).

**Figure 12 sensors-25-00635-f012:**
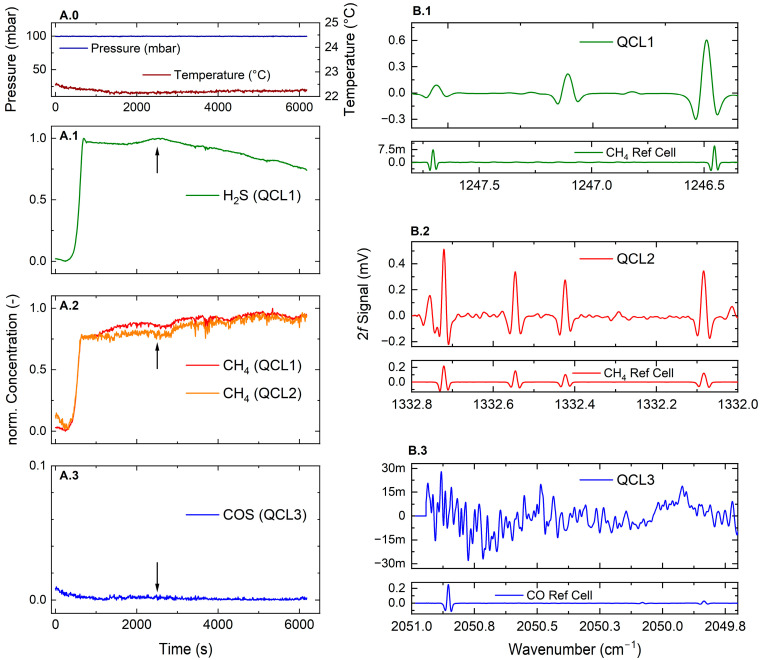
Time series measurements (**Panel A**) and detailed snap-shots of the spectral composition of each laser channel (**Panel B**): QCL1: H_2_S (green); QCL2: CH_4_ (red); QCL3: COS (blue).

**Table 1 sensors-25-00635-t001:** Laser parameters of the employed cw-QCLs.

#	Analyte	Manufacturer and SN	Emission Range (cm^−1^)	Operation Temperature Range (°C)	Operating Current Range (mA)	Modulation Current (mA)	Maximum Power (mW)
1	H_2_S	AdTech, Fairfax, VA, USA CM7-12-CI0329	1251–1246	15–35	280–570	0.5–10	220
2	CH_3_SH	Thorlabs, Newton, NJ, USA HZ-HHL-0170	1334.5–1330.5	15–35	110–290	0.2–5	160
3	COS	Alpes, Gisozi, Kigali Rwanda sbcw8283	2054.5–2049.5	15–35	110–150	0.1–2	30

**Table 2 sensors-25-00635-t002:** Parameters for the determination of the noise-equivalent absorption sensitivity (NEAS).

QCL	Analyte	Concentration	Averages	RMS Noise in Selected Region	SNR Peak	NEAS (cm^−1^ Hz^−0.5^)
1	H_2_S	25 ppmv	1	2.82 × 10^−4^	32.7	5.93 × 10^−9^
2	CH_3_SH	50 ppmv	1	2.21 × 10^−4^	54.6	4.65 × 10^−9^
3	COS	1.0 ppmv	1	2.49 × 10^−5^	442.3	5.24 × 10^−10^

## Data Availability

Data can be shared via personal request via email.

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
