# Peer review of "ATEX-Certified, FPGA-Based Three-Channel Quantum Cascade Laser Sensor for Sulfur Species Detection in Petrochemical Process Streams"

_sensors, 2025, doi:10.3390/s25030635_

Round 1
Reviewer 1 Report
Comments and Suggestions for Authors
This paper proposed a model of real-time gas sensor based on MIR optics, which was able to differentiate and quantify H2S, CH3SH and COS gases simultaneously, with a low detection limit and a high sensitivity. However, there are still some issues that need to be modified.
1. In the last part of the introduction, the optimal operating parameters for HDS or FCC are suggested to be provided.
2. The sensors in this paper were obtained by optimization on previous versions. There is no obvious comparative discussion of current sensors with relevant developments in the field. More rational and original explanations should be able to be reflected.
3. Figure 2: the lack of further discussion and analysis on the figure.
4. It is recommended to add literature published in recent years to exhibit the advantages of this sensor.
Author Response
Dear Ladies and Gentlemen,
Please find the revised and tracked changes in the supplied document - the final uploaded document reflects all changes made.
Sincerely

Reviewer 2 Report
Comments and Suggestions for Authors
The paper is focused on an interesting topic in the research field of multigas detection with optical sensors. I suggest some minor revisions before submission:
- Please, align the pictures with their captions
-Please, use subscripts for chemical species at row 136
-Please, please replace colon with a dot at row 187
-Please, use the subscript "0" for the angular frequency
-I suggest to show what happens for the H2S peaks in wavenumber tracking, since they are more distant and with different values of signal
-Please, make the label of figure 7 more visible
-Please, clarify in the text why it was used the wavenumber tracking if reference cells are employed in the sensor architecture
-In figure 9, it seems that there are fringes in the H2S and CH3SH signal. Please, comment and justify this behavior in the text.
-Please, use the subscripts for the analyte column ad the unit of measurement in table 2
-Please, highlight in the text if the stability in figure 10 was achieved with line locking or wavenumber tracking
-Please, clarify which wavenumber of the spectrum shown in Figure 11 B.3 was used to plot the time series of the COS values in Figure 11 A.3
Author Response

(The authors gave the same response as above.)

Reviewer 3 Report
Comments and Suggestions for Authors
In this work, a sensitive, selective, and industrially compatible gas sensor prototype is presented by utilizing three distributed feedback quantum cascade lasers (DFB-QCLs), and employing wavelength modulation spectroscopy (WMS) for the detection of hydrogen sulfide (H2S), methane (CH4), methyl mercaptan (CH3SH), and carbonyl sulfide (COS) in the spectral regions of 8.0 µm, 7.5 µm, and 4.9 µm, respectively. To demonstrate this detection technique, field-programmable gate array (FPGA) hard-ware is used for real-time signal generation, laser driving, signal processing and handling industrial communication protocols. However, how to achieve the simultaneous detection of three gases ? The technical principle is not clearly explained, Time division multiplexing or frequency division multiplexing ? and the related modulation parameters for individual laser ? Moreover, various spectroscopic sensing techniques have been successfully for multiple gases simultaneously detection, especially for off-axis integrated cavity output spectroscopy (Optics Communications 545:129731,2023), Dual-spectroscopy technique, and quartz crystal tuning fork enhanced laser spectroscopy. These previously reported gas sensing techniques can be discussed for comparison in the introduction. Comparing to three laser light sources used in this work, the mentioned multi-gas sensing techniques should have more advantages in view of cost, size and time resolution, etc.
Wavelength modulation spectroscopy with second harmonic detection (2f-WMS) is a well-known sensitive detction technique. However, the key technical problem of this technology is how to accurately calibrate the gas sensor system, especially in the field application process. For developing calibration-free WMS detection methods over the past decades, various novel methods or strategies for rapid inversion of gas concentration from WMS-2F spectrum have already reported, such as Multiple Linear Regression algorithm by using reference spectrum with known gas cylinder for field applications, WMS detection with direct absorption spectroscopy correction based dual-spectroscopy technique, self-calibrated 2f/1f wavelength modulation spectroscopy (ACS Sensors 5(11): 3607-3616,2020), as well as other detection methods mentioned references therein. These previously reported WMS signal processing methods have higher computational efficiency, and should be discussed in the introduction.
Author Response

(The authors gave the same response as above.)

Round 2
Reviewer 1 Report
Comments and Suggestions for Authors
All the issues I raised have revised well. Therefore, the revised manuscript can be accepted for publication.